# Synthesis and Properties of New Multilayer Chitosan@layered Double Hydroxide/Drug Loaded Phospholipid Bilayer Nanocomposite Bio-Hybrids

**DOI:** 10.3390/ma13163565

**Published:** 2020-08-12

**Authors:** Dan A. Lerner, Sylvie Bégu, Anne Aubert-Pouëssel, Ramona Polexe, Jean-Marie Devoisselle, Thierry Azaïs, Didier Tichit

**Affiliations:** 1ICGM, University of Montpellier, CNRS UMR 5253, ENSCM, 34296 Montpellier, France; sylvie.begu@enscm.fr (S.B.); anne.aubert@univ-montp1.fr (A.A.-P.); ramona_polexe@yahoo.com (R.P.); jean-marie.devoisselle@enscm.fr (J.-M.D.); 2Laboratoire de Chimie de la Matière Condensèe de Paris (LCMCP), Collège de France, Sorbonne Université, CNRS, F-75005 Paris, France; thierry.azais@sorbonne-universite.fr

**Keywords:** layered double hydroxide, phospholipid bilayer, chitosan, 17β-estradiol, bio-hybrid, drug delivery

## Abstract

A novel bio-hybrid drug delivery system was obtained involving a Mg/Al-NO_3_ layered double hydroxide (LDH) intercalated either with ibuprofenate anions (IBU) or a phospholipid bilayer (BL) containing a neutral drug, i.e., 17β-estradiol, and then embedded in chitosan beads. The combination of these components in a hierarchical structure led to synergistic effects investigated through characterization of the intermediates and the final bio-composites by XRD, TG, SEM, and TEM. That allowed determining the presence and yield of IBU and of BL in the interlayer space of LDH, and of the encapsulated LDH in the beads, as well as the morphology of the latter. Peculiar attention has been paid to the intercalation process of the BL for which all available data substantiate the hypothesis of a first interaction at the defect of the LDH, as well as on the interaction mode of these components. ^1^H, ^31^P and ^27^Al MAS-NMR studies allowed establishing that the intercalated BL is not homogeneous and likely formed patches. Release kinetics were performed for sodium ibuprofenate as well as for the association of 17β-estradiol within the negatively charged BL, each encapsulated in the LDH/chitosan hybrid materials. Such new bio-hybrids offer an interesting outlook into the pharmaceutical domain with the ability to be used as sustained release systems for a wide variety of anionic and, importantly, neutral drugs.

## 1. Introduction

Organic-inorganic hybrid materials formed by intercalation of anionic drugs into the interlayer spaces of layered double hydroxides (LDH) have recently gained much attention in the pharmaceutical domain due to several useful properties such as high drug loading, biocompatibility, enhanced cellular uptake, and delivery efficiency by anionic exchange or dissolution of the LDH matrix. LDH, also named anionic clays or hydrotalcite-like materials, although existing as natural minerals, are more commonly synthesized in the laboratory. They consist of brucite-like sheets based on octadedrally coordinated metallic cations (M^2+^ and M^3+^) surrounded by hydroxyl groups, held together by the presence of exchangeable anions (A^n-^) and a variable number of water molecules in the interlayer space. The ideal formula for an LDH is [M^2+^_1−x_ M^3+^_x_ (OH)_2_][(A^n−^)_x/n_ m H_2_O], in which x ranges between 0.2 and 0.33. Many pharmaceutically active compounds negatively charged at the physiological pH have been intercalated in LDH, forming two-component nanoparticles. This was the case for instance for anticancer drugs (camptothecin), anti-inflammatory drugs (fenbufen, ibuprofen), DNA, captopril, siRNA, as well as amino-acids and peptides [1,2,3,4,5,6,7]. This wealth of developments gave rise to the publication of many reviews of interest [8,9,10,11,12,13].

Despite several advantages, among which is the increase in the half-life and bioavailability of the encapsulated molecule, such drug-loaded LDH exhibited poor mechanical stability. Drug administration entails specific requirements, namely lower the toxicity, increase the efficiency of drug loading and release or improve the bio-distribution in different organs, goals rarely achieved with the two-component nanoparticles. So, in a more recent period, research was oriented towards the synthesis of more complex LDH-based nanoparticles. In particular, coating the drug-loaded LDH with various bio-polymers satisfied many of the above requirements, opening one of the most promising paths to better delivery systems. For instance, Yan et al. prepared a PEGylated phospholipid to externally coat LDH loaded with methotrexate and other drugs [14]. These nanoparticles could be internalized by cells and offered a good control of some cancer tumours. Zhanga et al. used ß-methasone intercalated in LDH [15]. The resulting nanohybrids were encapsulated in lecithin-cholesterol liposomes, which offered a good sustained release. Huang et al. prepared dextran magnetic LDH-fluorouracil particles containing an iron oxide core [16]. The latter was then externally coated by a lecithin-cholesterol bilayer, in which the magnetic properties were preserved. In an original study, Li et al. combining LDH and fulvic acid anchored on the surface through coagulation process prepared a multifunctional composite adsorbent that possesses the simultaneous ability for anion exchange and for toxic metal complexation [17].

Chitosan is a natural polysaccharide comprising copolymers of glucosamine and *N*-acetyl glucosamine usually obtained by partial de-acetylation of chitin from crustacean shells [18,19]. The latter reaction removes most of the acetyl groups from chitin and frees amine groups that can be used for further functionalization or charge-charge interactions as their pKa is ~6.5. Chitosan has been used in different fields, ranging from water waste management to medicine and biotechnology [20,21]. It becomes an interesting material in pharmaceutical applications due to its biodegradability, biocompatibility, and low toxicity [22,23,24]. The use of chitosan in novel drug delivery systems for peptide, gene delivery, as well as muco-adhesive device, oral enhancer or sensor for glucose or cholesterol has been reported in the literature [25,26,27,28,29,30].

LDH-chitosan nanocomposites have been used in a wide range of applications mainly as a drug delivery system. Generally, the anionic drug was loaded by intercalation between the LDH nanoplates, and the LDH-drug hybrid was coated with chitosan. However, in a few cases the nanocomposite morphology was that of an intimate mixture of individual LDH and chitosan components. Such composites generally increased the drug loading efficiency and the sustained release properties in comparison with pure LDH. They have been particularly investigated for bone regeneration [31,32,33]; treatment of ocular posterior segment and surface diseases of eyes [34,35,36]; control release of antibiotics like ciprofloxacin [37]; and photodynamic therapy due to the photosensitizer stabilization of a near-infrared fluorescent dye, indocyanine, in the interlayer of LDH [38]. LDH-chitosan nanocomposites based on a mixture of the two components show remarkable antibacterial activity [39,40]. On the other hand, intercalation of chitosan into LDH also gives rise to bio-nanocomposites exhibiting interesting antimicrobial activity [41]. However, in spite of several advantages of LDH-chitosan bio-hybrids, there is a great need to design systems exhibiting all their properties but also able to deliver neutral drugs of high pharmaceutical interest.

Here we first report on the preparation and characterization of beads of chitosan loaded with bio-hybrids formed by LDH exchanged with ibuprofen (IBU) in its anionic form. We then extended this approach to new drug-carrying nanocomposites made from beads of chitosan containing LDH nanoparticles previously intercalated with an anionic lipid bilayer (BL) (Scheme 1). The BL originates from drug-loaded liposomes that interact with LDH as previously shown by us [42]. The bilayer fills two functions: it solubilizes lipophilic neutral drugs that it will then release upon anionic exchange process. 17β-Estradiol (E) and 17β-estradiol acetate (A) were the neutral drugs used in the present work (See Scheme 1). In a parallel study, we carried the first NMR investigation of the interaction between an intercalated anionic bilayer and LDH using ^1^H, ^31^P and ^27^Al solid state MAS-NMR. This study allowed establishing that the intercalated BL is not homogeneous and likely formed patches, a fact that could affect the release of the drugs. Finally, some preliminary drug release experiments were carried out with this new material, at physiological pH conditions that triggered the process and comforted our original observation concerning the formation of new liposomes containing the drug in the release medium [42].

## 2. Materials and Methods

### 2.1. Preparation of the Mg/Al-NO_3_ LDH

The Mg/Al-NO_3_ LDH was synthesized by co-precipitation at increasing pH [43]. A solution containing 40 g Mg(NO_3_)_2_ 6H_2_O (99%, Fluka) (0.6 M) and 28 g Al(NO_3_)_3_ 9H_2_O (Fluka) (0.3 M) (Mg^2+^/Al^3+^ molar ratio of 2) dissolved in 250 mL deionized water was first prepared. To that solution poured into a beaker, a NaOH solution (2 M) was rapidly added until a pH value of 10 was reached. After complete precipitation, the resulting gel was stirred for 1 h. The precipitate was then repeatedly washed at 25 °C with distilled water, centrifuged, and finally dried by spray-drying (Spray Dryer B-290, Büchi Labortechnik, Essen, Germany).

### 2.2. Intercalation of the Drugs into Mg/Al-NO_3_ LDH

#### 2.2.1. Intercalation of Ionized Ibuprofen into Mg/Al-NO_3_ LDH

An amount of 1.28 g of ibuprofenate sodium dihydrate (Sigma-Aldrich, Lyon, France), corresponding to 1.11 g ibuprofenate ions (noted IBU), was dissolved and totally ionized in 100 mL of distilled water brought to a pH of 7 (ibuprofen acid has a pKa of about 4.5). This solution was used to prepare the LDH/ibuprofenate nanohybrid (LDH-IBU) by anionic exchange of NO_3_^−^ by IBU. For this purpose, 1 g of Mg/Al-NO_3_ was dispersed under vigorous stirring in the above solution. The initial amount of IBU corresponded to 1.31 times the anionic exchange capacity (AEC) of the LDH (AEC = 3.70 meq g^−1^). Using UV absorption spectroscopy, it was found that 18% of the IBU (197.5 10^−3^ g) remained in the supernatant at the end of the process. So about 82% of the total IBU was intercalated into the LDH. This corresponded to a total exchange of NO_3_^−^ ions by ibuprofenate ions. In a separate test, an aliquot of the sample was centrifuged and repeatedly washed in distilled water but no released IBU could be detected.

#### 2.2.2. Intercalation of the Drug Loaded Lipid Bilayer into Mg/Al-NO_3_ LDH

The LDH nanocomposites containing lipid bilayers (BL) in which lipophilic drugs were solubilized, were obtained in a two-step process. In the first one, self-assembled uni-lamellar anionic liposomes of dimyristoyl-phosphadityl-glycerol (DMPG, from Lipoïd, Ludwigshafen am Rhein, Germany) containing 8% (*w*/*w*) of the hormone 17β-estradiol (E) or 17β-estradiol acetate (A) (Sigma, Paris, France) were prepared according to the method of Bangham [44]. There were differences in the solubilisation of the drug E and its prodrug A, which is more hydrophobic, and this had to be taken into account [45,46,47]. After extrusion (extruder Lipex Biomembranes Inc., Vancouver, BC, Canada) through a set of polycarbonate filters (GE water and process technologies, San Jose, CA, USA) with mean pore size of 400, 200, and 100 nm, successively, the final suspension of small unilamellar liposomes contained 10 mg of lipids per mL of a 5 mM NaCl solution, with a pH adjusted at 7.4 using 0.1 M HCl. The size of the liposomes determined by DLS was a function of their hormone loading, varying from 58 to 80 nm for E and from 70 to 90 nm for A. In the second step, the drug-loaded liposomes intercalated themselves as guest phospholipid bilayers (noted BLE and BLA) via anionic exchange with the nitrate-compensating anions of the host LDH [3]. The intercalation involved the addition of 200 mg of LDH to 32 mL of the drug-loaded liposome suspension so that the expected quantity of intercalated drugs represented about 5% (w/w) in the final nano-biohybrid. It should be noted that when the liposomes were submitted to the extrusion process, a residual insoluble minor fraction of non-intercalated drug of about 10 wt% was retained on the filtration membrane. The intercalation was done at 37 °C, which is above the transition temperature of DMPG (23 °C), with a 48 h contact time in sealed flasks. The resulting solids were then separated by centrifugation for 5 min at 4500 rpm (Sigma 2K15, Bioblock Scientific, Illkirch, France), washed three times with distilled water and dried for 12 h at 40 °C. Thereafter, they will be labelled LDH-BLE and LDH-BLA, respectively, or LDH-BLE (A) when either E or A will be involved in the same conditions.

### 2.3. Preparation of LDH-loaded Chitosan Beads

#### 2.3.1. Preparation of Chitosan Solutions

An aqueous solution of chitosan (C) was obtained by dissolving 400 mg of C (Aldrich low molecular weight < 14% acetylation) in 20 mL of a solution of glacial acetic acid (Sigma-Aldrich, Lyon, France). The volume of glacial acetic acid was chosen to reach an equivalent amount of acid and NH_2_ functions. Total dissolution was obtained after stirring overnight at room temperature leading to a 2% (*w*/*w*) content in C.

#### 2.3.2. Preparation of the Chitosan Beads

In view of the many parameters involved in the formation of beads, their preparation was at first thoroughly studied with unloaded Mg/Al-NO_3_ LDH. Once the method to obtain these C-LDH beads was perfected, it was extended to the synthesis of LDH-IBU and LDH-BLE(A). For the latter, various amounts of LDH-IBU or LDH-BLE(A) were added to the solution of C. The resulting suspension was homogenised by vortex stirring. It was then added dropwise to a NaOH (2 M) solution through a 0.7-mm syringe needle. At this high pH, each drop gelified to form the final hybrid bead. The resulting new materials, C-LDH-IBU and C-LDH-BLE(A), respectively, were washed and their drying was tested by three different methods: in ambient air, supercritical CO_2_ treatment or lyophilisation. The two first methods only were found to keep the spherical shape of the particles but the supercritical drying was the only one to also maintain their size.

The supercritical CO_2_ drying process required an additional step of conversion of a hydrogel into an alcogel. In this process, the hydrogel was immersed in a series of successive ethanol-water baths (50 cm^3^) of increasing alcohol concentration (10, 30, 50, 70, 90, and 100%) for 15 min each. The chitosan beads were then dried in supercritical CO_2_ condition in a Polaron 3100, at 74 bar and 31.5 °C.

### 2.4. Characterization of the Materials

X-ray diffraction (XRD) patterns were recorded on a Bruker D8 Advance X-ray diffractometer (Bruker, Billerica, MA, USA) using Cu Kα radiation (λ = 1.542 Å, 40 kV and 50 mA).

TG-DTG experiments were carried out using a STA 6000 thermogravimetric analyzer (PerkinElmer, MA, USA) at a typical heating rate of 10 °C min^−1^ from room temperature up to 800 °C, in a nitrogen atmosphere (flow rate: 20 mL^.^min^−1^).

Transmission electron microscopy (TEM) was performed with a JEOL 1200 EX II (JEOL Europe SAS, Croissy sur Seine, France) apparatus, using an accelerating voltage of 200 kV. Powder samples were scraped from a spin-coated LDH-IBU or LDH-BLE(A) film under a microscope, diluted and dispersed in water and finally a drop was deposited onto uncoated copper TEM grids. Scanning Electron Microscopy (SEM) images were obtained using a Hitachi S-4800 microscope (Hitachi, Tokyo, Japan) after platinum metallization.

Viscosity was measured with a Rheomat RM 200 at room temperature.

IBU and E(A) concentrations were determined spectrophotometrically on a UV Lambda 35 (Perkin–Elmer). For the calibration curves, standard solutions (from 1mg mL^−1^ to 0.01 mg mL^−1^) were prepared in an aqueous solution at pH 10 for IBU and in spectroscopic grade methanol for E(A) (values read at the maximum of their UV-absorbance, in 10 mm quartz cell at a 281 nm (ε = 17600) for E and (ε = 16700) for A).

To determine the temperature of the phase transition of the phospholipid bilayer in its different environments and further to follow the release kinetics of the drug, the liposomes were loaded with the hydrophobic fluorescent probe 1,6-diphenyl-1,3,5-hexatriene (DPH) (Sigma-Aldrich, Lyon, France) together with the drug when required. The molar ratio of the probe to the lipids or (1-(4-trimethylammoniumphenyl)-6-Phenyl-1,3,5-Hexatriene p-Toluenesulfonate) (TMA-DPH) was set at 1/200.

Dynamic light scattering (DLS) measurements were performed at 25 °C on a NanoZS instrument (Malvern, England). The scattering measurements were done at a scattering angle of 173°.

^1^H and ^27^Al solid state NMR spectra were recorded on a AV700 Bruker spectrometer (Bruker, Billerica, MA, USA). Samples were packed in 1.3 mm zirconia rotors and spun at a spinning rate of 60 kHz. For one and two dimensional ^1^H-^27^Al CP MAS spectra, the Hartmann–Hahn condition was set for a low power irradiation of ^27^Al (RF = 5 kHz) in order to avoid any significant distortion of the quadrupolar line shape of the ^27^Al resonances. Recycle delays (RD) were set to 10 and 0.5 s for ^1^H and ^27^Al, respectively. ^31^P solid state NMR spectra were recorded on a AV300 Bruker spectrometer. Samples were packed in 4 mm zirconia rotors and spun at a spinning rate of 5–12 kHz. RD was set to 30 s. For the two dimensional ^27^Al-^31^P CP MAS spectra, the contact time was set to 2.5 ms. Chemical shifts were referenced (δ = 0 ppm) to tetramethylsilane, in 1M Al(NO_3_)_3_ aqueous solution, and H_3_PO_4_ 85 wt% for ^1^H, ^27^Al and ^31^P, respectively.

### 2.5. In Vitro Release of the Drug from the Nanohybrids

The drug release profiles of IBU were obtained from samples suspended in a PBS buffer pH 7.4 in sealed tubes clamped to a rotating wheel enclosed in an oven set at 37 °C. They were compared to the kinetics profiles of LDH-IBU in order to monitor the release from the C-LDH-IBU nanobiohybrids.

To monitor the release of E(A) from the C-LDH-BLE(A) nanocomposites, a small amount of C-LDH-BLE(A) beads was added to 15 mL of a phosphate buffer solution (150 mM, pH 7.4, 37 °C) and left under soft stirring. At selected times, 1 and 7 days, the supernatant was analyzed for its content in E(A) by UV absorption spectroscopy to control the amount of drug released. As previously observed, a fraction of the BL was also released or reconstituted [42]. The presence of a bilayerphase transition temperature after the release necessitated the use of a fluorescent probe since the drugs used were not fluorescent. To determine the phase transition temperature characteristic of drug-loaded liposomes (E, A, IBU), a 1 mM DPH solution was prepared in THF. One milliliter of the liposome suspension (blank or drug loaded) was mixed with 14 mL of 5 mM NaCl solution. DPH was added using a 1/200 DPH/liposome molar ratio (0.5% DPH moles based on the total number of moles) or 79 μL DPH solution for 15 mL of dilute DMPG liposomes. The preparation was preserved from light and stirred in a 37°C oven for 30 min to promote the loading of the DPH into the phospholipid bilayer. Emission anisotropry r was measured on a K2 spectrofluorimeter (ISS Inc., Champaign, IL, USA) equipped with quartz polarizers and temperature cell control (Excitation wavelength 360 nm, emission wavelength 430 nm).

## 3. Results and Discussion

### 3.1. LDH-Loaded Chitosan Beads

#### 3.1.1. Optimisation of the Preparation of the C-LDH Beads

C-LDH beads were obtained via a sol-gel process. The morphology and texture of the materials could be controlled by monitoring the viscosity of the gel and the final drying process. The chitosan hydrogel was easily shaped into beads by the stepwise dropping of an acidic solution of chitosan into an alkaline solution. The difficulties with beads preparation were related to the physico-chemical properties of chitosan (viscosity, pH dependence) [48,49]. For a pH value higher than 6.3 (neutral or alkaline pH), chitosan was stable but insoluble. At a pH lower than 6.3, the amino groups from the N-acetyl-glucosamine subunits of chitosan were protonated, hence positively charged, resulting in its solubilisation in water (polyelectrolyte effect). The solution of chitosan at pH 5 and the solid LDH were quickly mixed under vigorous stirring and the resulting suspension was added to a NaOH solution (2 M) leading to the hybrid precipitation and to the corresponding increase in pH and viscosity. Formation of stable C-LDH microspheres was determined using a set of samples with increasing amounts of LDH added to a 2% (*w*/*w*) solution of C at a pH of 5. These samples were listed as C-LDH(x) in which x represented the weight ratio (*w*/*w*) (Table 1).

The C-LDH(x) beads stability depended on the viscosity of the C solution. After the first addition of LDH, the viscosity of the C/LDH mixture (pH value 6.4) was about 120 mPa·s. Further addition of LDH increased the pH to 8.9 and the viscosity of the suspension reached 712 mPa·s for C-LDH(1.5). The mixture became more and more sluggish and finally solidified for C-LDH(3). The optimized beads, C-LDH(1), resulted from the mixing of C and LDH at a 1:1 weight ratio.

#### 3.1.2. Morphological and Structural Studies

All freshly prepared beads had a spherical shape with a diameter always close to 2 mm. The diameter shrank to 1 mm when the beads were left to dry in ambient conditions as shown on the photographs (Figure 1). C and C-LDH beads, dried using supercritical CO_2_, were examined by SEM and TEM. Micrographs of typical samples are presented in Figure 1 and Figure 2, respectively.

SEM micrographs of pure chitosan beads revealed a spherical shape with a rather smooth surface (Figure 1A) while their microtome section uncovered a fibrous structure for the core (Figure 1B). C-LDH beads displayed some porosity (Figure 1C) and a higher magnification revealed the presence of pseudo-hexagonal aggregates typical of LDH platelets in the shell (Figure 1D) mixed to the fibrillar structure of chitosan.

TEM micrographs revealed that pure C formed elongated fibers (Figure 2A). Quasi-spherical particles of LDH with diameter below 400 nm were identified in the core of C-LDH(1) beads (Figure 2B). Under higher magnification (Figure 2C), typical elongated LDH particles appeared. These results agreed with those obtained by SEM.

Previous structural studies by XRD showed that C adopted a twofold helical conformation resulting in asymmetric reflection due to the semi-crystalline state of the monosaccharide [50]. The XRD pattern of C displayed two sharp reflections at 10° and 20° (2θ) typical of chitosan and revealed its high crystallinity in agreement with previous studies (Figure 3e) [51]. The high crystallinity is due to the chitosan structure while hydroxyl groups could form strong intermolecular and intramolecular hydrogen bonds.

The pattern of freshly prepared Mg/Al-NO_3_ LDH (Figure 3a) showed two broad reflections with maxima around 10° and 20° (2θ) ascribed to (003) and (006) planes, respectively. Their position allowed calculating a d003-value of 0.84 nm in agreement with the presence in the interlayer space of nitrate anions provided by the Mg and Al precursor salts. The asymmetric reflections recorded in the center of the scanned range of the 2θ angles were due to non-basal reflections with turbostratic disorder. This sample exhibited low crystallinity as no hydrothermal treatment was performed on the gel after coprecipitation. Characteristic reflections of both C and LDH structures were found in the XRD patterns of the C-LDH(x) nanocomposites dried using supercritical CO_2_ [50].

The relative intensity of the (003) and (006) reflections increased with the amount of LDH introduced in the synthesis of these nanocomposites as seen when going from C-LDH(0.5) to C-LDH (2) (Figure 3b–d) [52]. Furthermore, compared to Mg/Al-NO_3_ LDH, these reflections were shifted toward higher 2θ values corresponding to a d003-value of 0.76 nm. This shrinkage of the basal reflection revealed that exchange of NO_3_^−^ by CO_3_^2−^ anions occurred upon drying using supercritical CO_2_. It allowed to clearly distinguish the (006) reflections of the LDH phase, which in the nitrate form were superimposed to that of C.

The TG profiles of the C-LDH(x) nanocomposites were compared to those of C and Mg/Al-NO_3_ LDH (Appendix A). The decomposition of C occurred in three steps. The first mass loss from 30 to 260 °C corresponding to 17% was assigned to the departure of water molecules. An abrupt mass loss of 18% occurred between 260 and 310 °C that was followed by a smooth and continuous mass loss up to 680 °C, both assigned to a depolymerisation reaction by decomposition of the amino or N-acetyl groups. The total mass loss reached 94%. The TG profile of Mg/Al-NO_3_ LDH was typical of this type of material with a mass loss in two steps. The first step from 30 to 200 °C corresponding to 13% of the original sample was generally attributed to removal of both water molecules physically adsorbed on the external surface of the crystallites and hydrogen bonded to the intercalated anions. The second step between 200 and 550 °C corresponding to a mass loss of 35% was due to the decomposition of the interlayer anionic species and to the dehydroxylation of the brucite-like sheets giving rise to the departure of water molecules. The total mass loss of 48% at 550 °C was close to the theoretical value of 51% corresponding to the decomposition of a Mg/Al-NO_3_ LDH (Mg/Al = 2) into MgO and Al_2_O_3_ oxides. The C-LDH(x) nanocomposites exhibited TG profiles with three different mass losses and general shapes close to that of C. The first mass loss from 30 to 260 °C in the range 14–18% was also followed by a second abrupt mass loss in the temperature range from 250 to 310 °C and corresponded to ~20–25%, while a third regular mass loss of 25–30% was recorded until 800 °C. The total mass losses of 70–80% were intermediate between those recorded for C and Mg/Al-NO_3_ LDH. A higher total mass loss of about 10% was observed for C-LDH(2) compared to C-LDH(1) and C-LDH(0.5) according to the higher LDH loading in the former sample. These results revealed the higher thermal stability of the LDH-containing beads than the bare C beads.

### 3.2. Loading of the Drugs in Liposomes

For the two estradiol types studied, size measured by DLS did not change linearly with loading amount. The liposomes ranged in size from 24 to 81 nm with poly-dispersity indexes from 0.22 to 0.38. For IBU, size did not change linearly either, with size varying between 24 and 71 nm but was more homogeneous with polydispersity indexes varying from 0.13 to 0.27. The mass yield in the exchange is detailed in Table 2.

With blank liposomes, an average of 55% of the material was recovered, that is, there was a partial loss of liposomes and LDH. When working with liposomes charged with, E, A, or IBU, this loss seemed to be accentuated. Also, clear differences in the behaviour of A, E and IBU were observed.

### 3.3. Loading of the Drugs into LDH

The next step of the work was to prepare LDH loaded with different drugs before its encapsulation in chitosan, as described above (Scheme 1). A first LDH-IBU hybrid material was obtained by anionic exchange of Mg/Al-NO_3_ LDH with ibuprofenate anions (IBU). The UV spectroscopic analysis of IBU content in the supernatant after exchange allowed determining that 40% of the initial NO_3_^-^ anions were removed.

A second type of material was prepared using a process that allowed the encapsulation of lipophilic neutral drugs. To trick LDH to accept the intercalation of a neutral drug, the latter was first solubilized in the lipid bilayer membrane (BL) of liposomes made from the negatively charged lipid DMPG. Here the drug was first solubilized in liposomes. The first intercalation of liposomes into LDH as a lipid bilayer was described by our group [42]. In the present study, the neutral lipophilic drugs chosen were 17β-estradiol (E, log P = 3.3) and 17β-estradiol acetate (A, log P = 3.7) for which the saturation solubility in the lipid bilayer was varied from 5 to 100 mg·mL^−1^ depending namely on the pH and ionic strength of the solution. That drug-loaded bilayer could penetrate the interlayer space of LDH by ion exchange [42]. It should be observed that such a lipid bilayer used as a vehicle to carry a drug has several interesting properties such as the surface charge, a nearly constant thickness and a dynamic flexibility, all of them necessary to penetrate the interlayer space of LDH and exchange the nitrate ions. This final material corresponded to LDH-BLE(A).

For the two hybrids LDH-BLE(A), the final mass yield of encapsulation was similar to that measured in liposomes (1.5%) but it was lower for IBU (about 1%).

The presence of IBU or BLE(A), in the interlayer space of the host Mg/Al-NO_3_ LDH, was confirmed by the XRD patterns of the LDH-IBU and LDH-BLE(A) nano bio-hybrid materials. After intercalation of IBU, the interlayer space of the host LDH increased from 0.84 to 2.10 nm. Similar results were obtained after intercalation of IBU into Mg/Al-Cl LDH by Ambrogi and al. [53]. The XRD patterns of Mg/Al-NO_3_ LDH and LDH-BLE were compared in Figure 4 and showed that upon exchange of BLE for NO_3_^−^ the intensities of the reflections located at low (1–7°) and high (7–70°) (2θ) ranges behaved in the opposite manner. Three harmonics at 1.74, 3.50 and 5.26° (2θ) corresponding to (003), (006) and (009) reflections with spacing-values of 5.05, 2.52 and 1.68 nm, respectively, were clearly identified in the low 2θ range. Concurrently, the intensity of the (003) and (006) reflections situated around 10° and 20° (2θ) assigned to interlayer spaces intercalated with NO_3_^-^, decreased. From the observed d-spacing of 5.05 nm and considering that the thickness of the brucite-like sheet is 0.48 nm, the interlayer space is 4.57 nm versus 4.16 nm found in a previous similar case [42]. It is noteworthy that 4.57 nm was only slightly smaller than the values of 4.79 and 4.75 nm reported for the DMPG bilayers below and above their transition temperature of 23.5 °C, respectively. This suggested a small tilting of the C-14 chains of the DMPG intercalated between the sheets of the LDH host. Other studies showed that hydrophilic and hydrophobic antibiotics anions caused an expansion of the interlayer space to 3.5 nm and dicetyl-phosphate to 5 nm [54,55]. So, the interlayer space observed in LDH-BLE was consistent with the intercalation of other organic anions of large size. It must be mentioned that very weak reflections were also recorded in LDH-BLE at 4.50° and 6.70° (2θ) corresponding to interlayer spaces of 19.60 and 13.18 nm, which can be assigned to (006) and (009) reflections of non-intercalated BL.

The TG profiles of the nano bio-hybrids were compared to those of the guest components IBU and BL and of the Mg/Al-NO_3_ LDH host already described (Figure 5). IBU was totally decomposed in one step between 180 and 260 °C. Less distinct mass loss steps were observed in LDH-IBU than in the host structure, the TG profile of the former sample being very similar to that previously reported [53]. Departure of hydration and intercalated water molecules leading to a mass loss of 15% at 250 °C was followed by a continuous mass loss of about 52% until 630 °C including the dehydroxylation of the LDH sheets and the decomposition of the intercalated IBU species. It is noteworthy that the thermal stability of the latter species was enhanced by more than 100 °C after intercalation. The total mass loss of 77% was close to the theoretical value corresponding to complete decomposition into oxides of Mg/Al-NO_3_ LDH totally exchanged with IBU. Decomposition of BL occurred in one step between 200 and 580 °C leading to a mass loss of about 80%. The TG profile of LDH-BLE had a similar shape to that of BL, however, with a lower total mass loss reaching 68% at 600 °C. Compared to Mg/Al-NO_3_ LDH, there was no first mass loss corresponding to departure of weakly bonded water molecules, which were not present due to steric hindrance by intercalated BL. One can also note that, contrary to IBU, the thermal stability of BL was not enhanced after intercalation. This accounted for a weaker interaction for BL than for IBU with LDH sheets, a fact that would be consistent with the presence of a BL with void patches.

In a previous publication, we tentatively proposed a crude mechanism to explain the way a liposome could slip between two brucite-like sheets [42]. What follows is a more detailed attempt to expound on the process, which could open new avenues of research. The hypothesis is made that the first interaction of a liposome with the LDH structure would be favored at the site of a defect. For instance, such a site could be one in which a sheet is not well aligned with its neighbours following an in-plane rotation or a distorted edge structure [56]. This interaction would somehow anchor the liposome by pinching its edge so that locally a transient double bilayer is formed. As we never got to see any intercalated double bilayer in spite of numerous efforts, we suggest that a flip-flop mechanism should occur to fuse the two layers into one that could then intercalate in the usual exchange process (Scheme 2). It is well known that at room temperature the flip-flop of lipids is too slow to account for the relatively fast process observed here for the liposome intercalation. However, recent modeling results point to the possibility of a large acceleration of the flip-flop of charged molecules (up to 10^4^) in a bilayer in special conditions [57]. More realistic is a mechanism in which one of the bilayers slides over the second one in a tank-tread-like motion, thanks to the presence of a thin-lubricating water film as observed by Rädler et al. [58]. It should be noted that to exchange the anions in a single interlayer space about 104 liposomes of an average diameter of 100 nm should intercalate [42]. This fact is in favour of the last mechanism proposed.

We could not find in the literature a previous solid state NMR study dealing with the interaction between the lipid bilayer and the LDH sheets in a LDH-BL hybrid. A first study was then undertaken to get complementary data on the organic-inorganic interactions and highlighted a modification of the phosphate head group environment due to the intercalation of the lipids inside the LDH interlayer space. First of all, ^31^P MAS NMR was used (Figure 6). LDH-BL possesses two different environments at 1.6 (site 1) and 0.2 (site 2) ppm in similar proportions (55 vs. 45%, respectively). Their resonances were slightly shifted from pure DMPG (δ(^31^P) = −1.3 ppm). Slow ^31^P MAS experiments (ν_MAS_ = 3 kHz) allowed the fitting of a spinning side band pattern and the extraction of the chemical shift anisotropy (CSA) parameters (Δ_CSA_ and η_CSA_) of the two sites. We found significant different values (Δ_CSA_ = −84 and −91 ppm, η_CSA_ = 0.65 and 0.55, for site 1 and 2, respectively) for the two sites confirming their different chemical environments. The rather high value of the Δ_CSA_ parameter was an indication of the rigidity of the phosphate polar head group of the lipid due to the strong electrostatic forces that stabilize the hybrid material.

Variable contact time ^1^H-^31^P CP MAS experiments (Appendix A), recorded at very high spinning speed (ν_MAS_ = 60 kHz) to minimize the ^1^H spin diffusion process, allowed the evaluation of the protonation of the phosphate group. The slow exponential increase of the ^31^P resonances intensities was characteristic of a non-protonated phosphate group and the absence of a POH moiety [59]. This result implied that the phosphate head group was not protonated, i.e., it was still negatively charged in the bilayers, leading to electrostatic interactions with the positively-charged LDH sheets.

Insight into the local interaction between the lipids and the LDH sheets could be obtained through ^27^Al solid state NMR experiments. The ^27^Al MAS spectrum of LDH-BL and pure LDH is depicted in Figure 7. Three different ^27^Al environments (A, B and C) were detected and confirmed by means of ^27^Al MQMAS experiment (Appendix A) among which the C resonance seems to correspond to an LDH domain in which the lipid was not intercalated.

The problem of LDH-BL intercalation can be tackled down through ^1^H-^27^Al CP MAS experiments. One dimensional ^1^H-^27^Al CP MAS spectra for various contact times are presented in Appendix A. A, B and C sites were observed, which meant that the low level of rf irradiation on the ^27^Al did not lead to any significant distortion of the possible quadrupolar line shape of the ^27^Al resonances. Thus, 2D ^1^H-^27^Al HetCor spectra recorded for three different contact times (400 µs, 2 and 10 ms) could safely be analysed (Figure 8). The comparison of the extracted ^1^H slices of the A, B and C sites and their comparison with ^1^H MAS spectra of LDH-BL and pure LDH was particularly informative. While the ^1^H MAS spectrum of the pure LDH exhibited a main resonance around 4 ppm, characteristic of Mg_2_AlOH groups from the LDH sheets [60,61], the ^1^H MAS spectrum of LDH-BL exhibited additional resonances coming from the lipid. There was particularly a strong resonance around 1–2 ppm, corresponding to the alkyl chains, and a broad and weak resonance around 6–8 ppm, corresponding to the OH from the glycerol moiety of the polar head group.

At short contact time (400 µs), the three ^27^Al sites correlated principally with the AlOH from the LDH sheet as expected. At moderate contact time (2 ms), the OH from the glycerol moiety and CH from the aliphatic chains started to be observed even if the corresponding resonances were weaker for the C ^27^Al site, indicating a longer distance between this aluminium site and the lipids, consistently with the ^27^Al MAS NMR spectrum. Finally, increasing the contact time to 10 ms, the correlations with the aliphatic chain were now prominent for the three aluminium sites.

From this series of experiments, it can be concluded that BL was intercalated between the LDH sheets but probably small patches without lipids were present. In particular, the correlation of the aluminium site C with BL proved that the hybrid LDH-BL material was not formed from independent particles with and without intercalated lipids but rather with particles where the intercalation is not homogenous and consequently, possessing domains with intercalated and non-intercalated BL. It is important to note that this result is in line with our experimental observations detailed above.

Up to here, the existence of two different ^31^P environments for the intercalated lipids was still unclear but a 2D ^27^Al-^31^P HetCor experiment could reveal the proximities between the lipids and the aluminium sites from the LDH. The 2D ^27^Al-^31^P HetCor spectrum of LDH-BL is depicted in Appendix A and exhibited an asymmetric line shape where the aluminium A site seemed to correlate with the phosphate at low chemical shift (site 2), whereas the B site correlated with phosphate at higher chemical shift (site 1). The origin of these two different environments both for the lipids and the ^27^Al atoms from the LDH sheets remains unclear, but could correspond to two different modes of interaction between the lipid headgroup and LDH layer.

### 3.4. Drug-loaded LDH in Chitosan Beads

The morphology of the drug-loaded C-LDH beads was identical to that of the C-LDH unloaded hybrids. The general shape of the XRD pattern for C-LDH-BLE beads (Figure 9c) was different from that of LDH-BLE (Figure 4b) particularly in the 2θ range from 10° to 25° where the peaks due to the phospholipid bilayers (BL) became less intense and merged with those of C giving a broad peak extending from 17° to 25° (2θ). Two straight and intense peaks were recorded at 4.46° and 6.62° (2θ) corresponding to the interlayer space of 1.98 and 1.33 nm, respectively, assigned previously to un-intercalated phospholipid bilayers in LDH-BLE (Figure 4b). The peaks corresponding to LDH-BLE became very weak in C-LDH-BLE. These results showed that both well-dispersed LDH-BLE and an amount of un-intercalated BLE were present in the C-LDH-BLE beads. The TG profiles of the hybrids C-HDL and C-HDL-IBU did not show significant differences.

### 3.5. Drug Release

As the fluorescent probe DPH used here has about the same size and weight as the drugs tested it should not modify the dynamic properties of the liposomes, especially since it has no charge and is used at a much lower concentration. DPH introduced at different loadings into blank liposomes induced changes in their average size from 60 to 80 nm at low concentrations, with a polydispersity index stable at PI = 0.2. The phase transition was observed at 26–27 °C. Given the strong influence of the loading for DPMG liposomes, a value of 0.5% per 200 molecules of DPMG was chosen for the content in DPH.

The study of the fluorescence anisotropy r following the co-loading of DPH in C-LDH-BLE allows the determination of the phase transition temperature (Tc) characteristic of the organized DMPG bilayer. Tc is measured experimentally at 23 °C.

The presence of estradiol E did not alter the curve profile or the Tc and anisotropy values with an initial r at 15 °C equal to 0.32 and a final r (40 °C) at 0.05. The presence of IBU changes the profile of the curve, and the values of Tc and anisotropy with an initial **r** at 16 °C equal to 0.32 and a final r (40 °C) at 0.06 +/−0.01.

The stability of the drug-loaded C-LDH beads was followed up to 3 months in a pH 7.4 PBS solution at 25 °C, and the results indicated that the beads were stable over that time span.

Preliminary tests revealed a slow release of IBU from simple LDH-IBU particles as well as a total release of IBU loaded in C alone in 2 h (at pH 2.2 and 7). In C-LDH-IBU, the drug release mechanism from the intercalated compounds occurred by exchange between IBU and the buffer phosphate ions diffusing through the net of the polymer chains. A pronounced burst was observed during the first 20 min and was followed by a slow release (Figure 10). This fast step was probably correlated with IBU molecules that were first released from LDH adsorbed on the surface of the LDH-IBU particles. In the next step, the release kinetics turned into a constant rate of efflux controlled by the drug low rate of escape from the LDH particles. Therefore, this system had the potential for the sustained release of IBU.

Release kinetic studies from C-LDH-BLE were also carried out in different release media (pH, nature of ions) to evaluate the potential for sustained release of the beads. The kinetics for the release from C-LDH-BLE were much slower and the amount of E released after 7 days was calculated to represent 0.03% of the total load (data not presented). It must be noted that the presence of a slight amount of cations such as Na^+^ provided by the liposome solution can influence the ionic strength and the drug delivery process [62]. The influence of such a parameter will be examined in future studies dealing with these new delivery systems.

## 4. Conclusions

These new materials present further advantages as they result from green synthesis conditions (without solvent) and allow the encapsulation of poorly water-soluble drugs, even if they carry no charge, due to their initial loading in DMPG liposomes.

In summary, the present work describes a novel drug delivery system based on LDH loaded with the drug and trapped into chitosan beads. It was found that neutral and anionic drugs could be encapsulated in this system and then released from the polymeric beads. The latter were kept intact after the release step. The encapsulation efficiency of LDH-BLE or LDH-IBU in the chitosan beads was about 100%. The encapsulation of LDH did not significantly change the overall particle size, and on their surface morphology nano spheres were observed. The presence of LDH on the surface of the beads was observed (by SEM). The in vitro release of the drugs from the chitosan beads could be controlled by adjusting the composition of the loaded C-LDH beads and the encapsulation process parameters. Such a novel controlled release system integrates the advantages and avoids the disadvantages of formulations based on polymer or LDH alone. It has a strong potential for drug delivery including that of poorly water-soluble and neutral drugs. Much work was done, however, if one considers the large number of parameters involved, a challenging path is opened to further improvements namely in the intercalation step of the drug-loaded liposomes in the LDH nanoparticles, the control of the LDH nanoparticles morphology, and the state and amount of chitosan.

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
