# Peer review of "Synthesis and Properties of New Multilayer Chitosan@layered Double Hydroxide/Drug Loaded Phospholipid Bilayer Nanocomposite Bio-Hybrids"

_materials, 2020, doi:10.3390/ma13163565_

Round 1

Reviewer 1 Report

In this manuscript, Lerner et al. designed a novel bio-hybrid drug delivery system was obtained involving a Mg/Al-NO3 layered double hydroxide (LDH) intercalated either with ibuprofenate anions or phospholipid bilayer containing neutral drug embedded in chitosan beads. Overall, the manuscript is well written, especially the materials and method section is very clear for the readers to duplicate their experiments. The work is interesting and might be useful for being a platform for drug delivery study. There are still numerous concerns with the experimental data, and description which should be addressed to improve the quality of the findings and conclusions.

  1. Since LDH is not a new type of material for biomedical applications, the author should stand out the advantages of this study and briefly discuss the novelty compared with other studies in the introduction.
  2. It will be easier for readers to follow if the authors can directly label the diffraction peaks with Miller indices in the XRD figure (Figure 3). If possible, the author should include the peaks of the standard sample from the JCPDS card. This is the same for Figure 4 and Figure 9.
  3. About the drug release studies, the data should start from 0 min. The y-axis shouldn’t be in the log scale. Could the author explain why the release efficiency increased and decreased over time for C-LDH-IBU group? That doesn’t make sense. The data should also include error bars to show that the experiments could be reproducible.
  4. If the author can provide simple in vitro studies, that will substantially prove the finding and enhance the quality of this study.

Author Response

In this manuscript, Lerner et al. designed a novel bio-hybrid drug delivery system was obtained involving a Mg/Al-NO3 layered double hydroxide (LDH) intercalated either with ibuprofenate anions or phospholipid bilayer containing neutral drug embedded in chitosan beads. Overall, the manuscript is well written, especially the materials and method section is very clear for the readers to duplicate their experiments. The work is interesting and might be useful for being a platform for drug delivery study. There are still numerous concerns with the experimental data, and description which should be addressed to improve the quality of the findings and conclusions.

1) Since LDH is not a new type of material for biomedical applications, the author should stand out the advantages of this study and briefly discuss the novelty compared with other studies in the introduction..

We have added in the introduction a new paragraph describing different types of chitosan-LDH nanocomposites studied these last five years and their main applications (bone regeneration, treatment of ocular diseases, antibiotic control release, antibacterial treatment…). We also emphasize the novelty of our approach involving intercalation into LDH of phospholipid loaded with neutral drug able for the first time to achieve delivery of such type of drug with at the same time all the advantages provided by the chitosan-LDH composites.
The following paragraph has been introduced (p 2 lines 73 - 86) :

« LDH-chitosan nanocomposites have been used in a wide range of applications mainly as drug delivery system. Generally, the anionic drug was loaded by intercalation between the LDH nanoplates and the LDH-drug hybrid was coated with chitosan. However, in a few cases the nanocomposite morphology was that of an intimate mixture of individual LDH and chitosan components. Such composites generally increased the drug loading efficiency and the sustained release properties in comparison with pure LDH. They have been particularly investigated for bone regeneration [31-33], treatment of ocular posterior segment and surface diseases of eyes [34-36], control release of antibiotics like ciprofloxacin [37], photodynamic therapy due to the photosensitizer stabilization of a near-infrared fluorescent dye, indocyanine, in the interlayer of LDH [38]. LDH-chitosan nanocomposites based on a mixture of the two components show remarkable antibacterial activity [39,40]. On the other hand, intercalation of chitosan into LDH also gives rise to bio-nanocomposites exhibiting interesting antimicrobial activity [41].  However, in spite of several advantages of LDH-chitosan bio-hybrids there is a great need to design systems exhibiting all these properties but also able to deliver neutral drugs of high pharmaceutical interest.»

2) It will be easier for readers to follow if the authors can directly label the diffraction peaks with Miller indices in the XRD figure (Figure 3). If possible, the author should include the peaks of the standard sample from the JCPDS card. This is the same for Figure 4 and Figure 9.

The Miller indexes have been introduced in Figures 3, 4 and 9. We apologize because it was not possible to introduce the peaks of the standard sample from JCPDS card. However, we think the assignement of the peaks appears clearly now.

3) About the drug release studies, the data should start from 0 min. The y-axis shouldn’t be in the log scale. Could the author explain why the release efficiency increased and decreased over time for C-LDH-IBU group? That doesn’t make sense. The data should also include error bars to show that the experiments could be reproducible.

Following the different points underlined by the reviewer, Figure 10 has been redrawn. The data now start from 0 min and error bars (~15%) have been introduced.

We have corrected the caption of Figure 10 underlining that it is a semi-log plot.

About the variation of release efficiency in the first 30 min for C-LDH-IBU we think that the explanation has been given in the following sentence already present in the text (p 16 lines 545 – 547) :

 « A pronounced burst was observed during the first 20 min and was followed by a slow release (Figure 10). This fast step was probably correlated with IBU molecules that were first released from LDH adsorbed on the surface of the LDH-IBU particles.”

4) If the author can provide simple in vitro studies, that will substantially prove the finding and enhance the quality of this study.

We have commented (page 17 lines 556 - 559) the release kinetics for C-LDH-BLE nanocomposite and we report that the amount of E released after 7 days represents 0.03% of the total load showing that it is ery slow. In the frame of this preliminary study we have not obtained more data. The release kinetic studies will be more extensively developed in our future work due to the great potential exhibited by the chitosan/LDH/ phospholipid bio-hybrids.

Reviewer 2 Report

Review of the Manuscript entitled:

Synthesis and properties of new multilayer 2 chitosan@layered double hydroxide/drug loaded 3 phospholipid bilayer nanocomposite bio-hybrids

The manuscript of Dan A. Lerner et al. presents an experimental investigation on a bio-hybrid drug delivery system obtained involving a Mg/Al-NO3 layered double hydroxide (LDH) intercalated either with ibuprofenate anions (IBU) or phospholipid bilayer (BL) containing neutral drug, 17β-estradiol, and then embedded in chitosan beads. The authors evidence that the manufactured nanocarriers (obtained from green synthesis conditions) resulted in a highly efficacious nanodelivery system, as they allow the encapsulation of poorly water-soluble drugs with optimal stability and a sustained release profile.

The development of new nano-formulations incorporating the drug molecules into biocompatible nanocarriers has great potential for improving efficacy and safety in drug delivery application. It also attracts increasing attention as it gives new insight into the main interactions and assembly processes that involve drug delivery processes with smart nanoparticles.

Most of the experimental description and interpretation are clearly presented and understandable in its present version. The paper only needs some major improvement (minor revision) before being suitable for publication.

The authors might take into account some of the remarks below:

[1] - At page 4 – line 122, the authors state that “After extrusion through a set of polycarbonate filters, the final suspension of small unilamellar liposomes 123 contained 10 mg of lipids per mL of a 5 mM NaCl solution, with a pH adjusted at 7.4 using 0.1 M 124 HCl.  The size of the liposomes determined by DLS was a function of their loading in hormone, varying from 58 to 80 nm for E and from 70 to 90 nm for A.

Please specify better the extrusion procedure performed. Which pore size of the filter was used ?. How the final size of the extruded liposomes were measured (by light scattering measurement or SEM/TEM ?).

Moreover, generally by means of the extrusion process, together the unilamellar liposomes, a small percentage of multilamellar liposomes are generally formed. It would have been advisable to carry out a control (just after the extrrusion) through SAXS or SEM measurements, to  about the final results (in terms of liposome lamellarity.

[2] - At page 4 – line 130, the authors state that “…It should be noted that when the liposomes were submitted to the extrusion process, a residual insoluble minor fraction of non-intercalated drug was retained on the filtration membrane.

Please add a comment about a possible estimate of the quantity of drugs excluded from inclusion.

[3] - At page 10 – line 318 the authors state that “…For the two estradiol types studied, size did not change linearly with loading amount. The liposomes ranged in size from 24 to 81 nm with poly-dispersity indexes from 0.22 to 0.38.

Please specify which technique have been employed (SEM/TEM or dynamic light scattering (DLS)).

Actually it would be better to compare the combination of two techniques.

[3] - At page 13 – line 409 the author suggest that “…a flip-flop mechanism should occur to fusion the two layers in one that could then intercalate in the usual  exchange process.

Please associate a drawing (or a graphic scheme) in order to better explain the hypothesized mechanism.

Minor Revision

At page 13 – line 562 (Conclusion Section) the authors repeat the authors repeat the following sentence twice …“It has a strong potential for drug 562 delivery including that of poorly water soluble and neutral drugs

Finally, English should be carefully revised throughout the manuscript.

Author Response

The manuscript of Dan A. Lerner et al. presents an experimental investigation on a bio-hybrid drug delivery system obtained involving a Mg/Al-NO3 layered double hydroxide (LDH) intercalated either with ibuprofenate anions (IBU) or phospholipid bilayer (BL) containing neutral drug, 17β-estradiol, and then embedded in chitosan beads. The authors evidence that the manufactured nanocarriers (obtained from green synthesis conditions) resulted in a highly efficacious nanodelivery system, as they allow the encapsulation of poorly water-soluble drugs with optimal stability and a sustained release profile.

The development of new nano-formulations incorporating the drug molecules into biocompatible nanocarriers has great potential for improving efficacy and safety in drug delivery application. It also attracts increasing attention as it gives new insight into the main interactions and assembly processes that involve drug delivery processes with smart nanoparticles.

Most of the experimental description and interpretation are clearly presented and understandable in its present version. The paper only needs some major improvement (minor revision) before being suitable for publication.

The authors might take into account some of the remarks below:

[1] - At page 4 – line 122, the authors state that “After extrusion through a set of polycarbonate filters, the final suspension of small unilamellar liposomes 123 contained 10 mg of lipids per mL of a 5 mM NaCl solution, with a pH adjusted at 7.4 using 0.1 M 124 HCl.  The size of the liposomes determined by DLS was a function of their loading in hormone, varying from 58 to 80 nm for E and from 70 to 90 nm for A.
Please specify better the extrusion procedure performed. Which pore size of the filter was used ?. How the final size of the extruded liposomes were measured (by light scattering measurement or SEM/TEM ?).

We have considered the suggestion of the reviewer and the sentence has been modified. We have indeed added the type of extruder and the pore size of the three successive filters used for the extrusion. Therefore the new sentence is (p 4 lines 136 – 140):

 “After extrusion (extruder Lipex Biomembranes Inc., Canada) through a set of polycarbonate filters (GE water and process technologies, US) with mean pore size of 400, 200, and 100 nm, successively, the final suspension of small unilamellar liposomes contained 10 mg of lipids per mL of a 5 mM NaCl solution, with a pH adjusted at 7.4 using 0.1 M HCl.”

As already indicated in the second sentence the final size of the liposomes has been determined using DLS. The experiments were performed at 25 °C on a NanoZS instrument (Malvern).

Moreover, generally by means of the extrusion process, together the unilamellar liposomes, a small percentage of multilamellar liposomes are generally formed. It would have been advisable to carry out a control (just after the extrrusion) through SAXS or SEM measurements, to about the final results (in terms of liposome lamellarity.

The reviewer is absolutely right and we used SAXS in similar cases and in studies of micelles of polymers, but,not in the present case as the SAXS machine is 400 Km away from our lab. and this is a problem. We used a recent DLS instrument and analyzed the polydispersity indices and intensities of the samples, and in these conditions it is difficult to get meaningful values.

[2] - At page 4 – line 130, the authors state that “…It should be noted that when the liposomes were submitted to the extrusion process, a residual insoluble minor fraction of non-intercalated drug was retained on the filtration membrane.
Please add a comment about a possible estimate of the quantity of drugs excluded from inclusion.

The quantity of drugs excluded from inclusion in BL was estimated to about 10% only. This value has been added to the sentence (p 4 line 146 - 148) :

« It should be noted that when the liposomes were submitted to the extrusion process, a residual insoluble minor fraction of non-intercalated drug of about 10 wt% was retained on the filtration membrane.”

[3] - At page 10 – line 318 the authors state that “…For the two estradiol types studied, size did not change linearly with loading amount. The liposomes ranged in size from 24 to 81 nm with poly-dispersity indexes from 0.22 to 0.38.
Please specify which technique have been employed (SEM/TEM or dynamic light scattering (DLS)).

The reviewer is right and the technique used must be indicated. We have specified that the measured has been done by DLS. The new sentence (p 9 lines 329 - 331) in the revised version is :

 « For the two estradiol types studied, size measured by DLS did not change linearly with loading amount. The liposomes ranged in size from 24 to 81 nm with poly-dispersity indexes from 0.22 to 0.38.”

Actually it would be better to compare the combination of two techniques.

We were not able to use both DLS and SEM/TEM in these experiments particularly.  It is not easy to prepare the samples for the microscopic analyses without affecting the nature of the samples.

[3] - At page 13 – line 409 the author suggest that “…a flip-flop mechanism should occur to fusion the two layers in one that could then intercalate in the usual  exchange process.
Please associate a drawing (or a graphic scheme) in order to better explain the hypothesized mechanism
.

We thank the reviewer for this suggestion because the mechanism is indeed not obvious. A simplified graphic scheme has been added (Scheme 2 at p 12 line 429)

Minor Revision

At page 13 – line 562 (Conclusion Section) the authors repeat the authors repeat the following sentence twice …“It has a strong potential for drug 562 delivery including that of poorly water soluble and neutral drugs.

The mistake has been corrected.

Finally, English should be carefully revised throughout the manuscript.

We have carefully read the manuscript and several changes have been performed in order to improve the English.

Reviewer 3 Report

In the manuscript by Lerner et al, the authors reported the preparation, structural characterization and drug delivery features of LDH-chitosan hybrid composites. Indeed, LDH materials are of great promise as delivery agents in biomedical treatments due to their biocompatibility, ease of synthesis and advantageous lamellar structure, however, their formulation must be precise to achieve the desired efficiency. In addition, polysaccharide derivatives have proven as sufficient components in composites for medical applications, they usually provide excellent structural and colloid stability for these delivery vehicles. Therefore, the topic is hot and the results should attract considerable attention in the academic and industrial research communities. The primary experimental results adequately support the conclusions and the text is well written, it is easy to follow. I recommend publication in the journal Materials, provided the following minor issues are addressed in a revision process.

(i) The structural characterization is solid, but one may wonder about the tendency in charging features during the development of the hybrid materials. Do intercalation of drugs or lipid bilayer change the overall charge of the delivery particles? Are any of the sodium ions are co-intercalated with the lipid? Such charging features often influence the delivery processes (see Wu et al., Nanomaterials 2019, 9, 159 for instance), so it deserves some discussions.

(ii) I did not find information on the accuracy of the quantitative data. For example, Tables 1 and 2 contains numerous values without indicating their errors. Please comment on this issue and give at least the average precision of the methods.

(iii) There are large number of studies dealing with LDH-polysaccharide complexes for delivery purposes in the literature. I recommend extending the general discussion of the topic and the reference list with relevant papers. Two suggestions: Pavlovic et al., J. Colloid Interface Sci. 2018, 524, 114-121 and Chen et al., Nanoscale 2017, 9, 6765-6776.

Author Response

In the manuscript by Lerner et al, the authors reported the preparation, structural characterization and drug delivery features of LDH-chitosan hybrid composites. Indeed, LDH materials are of great promise as delivery agents in biomedical treatments due to their biocompatibility, ease of synthesis and advantageous lamellar structure, however, their formulation must be precise to achieve the desired efficiency. In addition, polysaccharide derivatives have proven as sufficient components in composites for medical applications, they usually provide excellent structural and colloid stability for these delivery vehicles. Therefore, the topic is hot and the results should attract considerable attention in the academic and industrial research communities. The primary experimental results adequately support the conclusions and the text is well written, it is easy to follow. I recommend publication in the journal Materials, provided the following minor issues are addressed in a revision process.

(i) The structural characterization is solid, but one may wonder about the tendency in charging features during the development of the hybrid materials. Do intercalation of drugs or lipid bilayer change the overall charge of the delivery particles? Are any of the sodium ions are co-intercalated with the lipid? Such charging features often influence the delivery processes (see Wu et al., Nanomaterials 2019, 9, 159 for instance), so it deserves some discussions.

It is a very interesting comment.  Normally the delivery particles must be neutral because the intercalation of liposomes and/or drug with negative charge into the LDH interlayer must compensate the positive charge of the brucite like layers. The embedding in chitosan must not change the global charge. However, we have not done zeta potential experiments in order to check this hypothesis. This is one type of characterization which is planned for future developments of these systems. The present study was mainly a well tester proof of concept.
The intercalation of liposomes into LDH is achieved by anionic exchange of the nitrate compensating anions with the drug-loaded liposome suspension containing NaCl and HCl to adjust the final pH to 7.4 as described in the Experimental section. Therefore, Na+ cations can be present in the C-LDH-BL particles either on the external surface or trapped between the patches of BL in the LDH interlayer space. Similar feature has been reported by us for intercalation of MO in MgAl-LDH [G. Darmograi et al., J. Phys. Chem. C, 2015, 119, 23388]. During drug delivery, the presence of Na+ can increase the ionic strength and for exemple modify cellular uptake due to the increase of osmotic pressure [Wu et al., Nanomaterials, 2019, 9, 159]. Unfortunately, we have not achieved chemical analysis of residual Na present in our samples. But we have considered the remark of the reviewer and we have added the following sentence (p 17 lines 559 - 562):

 « It must be noted that presence of slight amount of cations such as Na+ provided by the liposome solution can influence the ionic strength and the drug delivery process [62]. The influence of such parameter will be examined in future studies dealing with these new delivery systems.”

(ii) I did not find information on the accuracy of the quantitative data. For example, Tables 1 and 2 contains numerous values without indicating their errors. Please comment on this issue and give at least the average precision of the methods.

The average precision (± 5 wt%) on the amount of drug present in BL has been added in Table 2.

(iii) There are large number of studies dealing with LDH-polysaccharide complexes for delivery purposes in the literature. I recommend extending the general discussion of the topic and the reference list with relevant papers. Two suggestions: Pavlovic et al., J. Colloid Interface Sci. 2018, 524, 114-121 and Chen et al., Nanoscale 2017, 9, 6765-6776.

We totally agree with this remark of the reviewer. A reference to several relevant papers dealing particularly with LDH/chitosan nanocomposites was lacking in our original manuscript.  In the introduction of the revised version of the manuscript we have added a paragraph containing 11 references dealing with different types of LDH/chitosan assemblies emphasizing on their application fields (bone regeneration, treatment of ocular diseases, control release of antibiotics, antibacterial activity..). We think that this allows to highlight that our study adress a relevant topic.

The following paragraph has been introduced (p 2 lines 73 - 86) :

« LDH-chitosan nanocomposites have been used in a wide range of applications mainly as drug delivery system. Generally, the anionic drug was loaded by intercalation between the LDH nanoplates and the LDH-drug hybrid was coated with chitosan. However, in a few cases the nanocomposite morphology was that of an intimate mixture of individual LDH and chitosan components. Such composites generally increased the drug loading efficiency and the sustained release properties in comparison with pure LDH. They have been particularly investigated for bone regeneration [31-33], treatment of ocular posterior segment and surface diseases of eyes [34-36], control release of antibiotics like ciprofloxacin [37], photodynamic therapy due to the photosensitizer stabilization of a near-infrared fluorescent dye, indocyanine, in the interlayer of LDH [38]. LDH-chitosan nanocomposites based on a mixture of the two components show remarkable antibacterial activity [39,40]. On the other hand, intercalation of chitosan into LDH also gives rise to bio-nanocomposites exhibiting interesting antimicrobial activity [41].  However, in spite of several advantages of LDH-chitosan bio-hybrids there is a great need to design systems exhibiting all thèse properties but also able to deliver neutral drugs  of high pharmaceutical interest. »